# A Lightweight Pavement Defect Detection Algorithm Integrating Perception Enhancement and Feature Optimization

**DOI:** 10.3390/s25144443

**Published:** 2025-07-17

**Authors:** Xiang Zhang, Xiaopeng Wang, Zhuorang Yang

**Affiliations:** School of Electronics and Information Engineering, Lanzhou Jiaotong University, Lanzhou 730070, China; 13273412563@163.com (X.Z.); 15002554467@163.com (Z.Y.)

**Keywords:** pavement defect detection, receptive field-guided attention, lightweight, detail-enhanced convolution, dynamic upsampling method

## Abstract

To address the current issue of large computations and the difficulty in balancing model complexity and detection accuracy in pavement defect detection models, a lightweight pavement defect detection algorithm, PGS-YOLO, is proposed based on YOLOv8, which integrates perception enhancement and feature optimization. The algorithm first designs the Receptive-Field Convolutional Block Attention Module Convolution (RFCBAMConv) and the Receptive-Field Convolutional Block Attention Module C2f-RFCBAM, based on which we construct an efficient Perception Enhanced Feature Extraction Network (PEFNet) that enhances multi-scale feature extraction capability by dynamically adjusting the receptive field. Secondly, the dynamic upsampling module DySample is introduced into the efficient feature pyramid, constructing a new feature fusion pyramid (Generalized Dynamic Sampling Feature Pyramid Network, GDSFPN) to optimize the multi-scale feature fusion effect. In addition, a shared detail-enhanced convolution lightweight detection head (SDCLD) was designed, which significantly reduces the model’s parameters and computation while improving localization and classification performance. Finally, Wise-IoU was introduced to optimize the training performance and detection accuracy of the model. Experimental results show that PGS-YOLO increases mAP50 by 2.8% and 2.9% on the complete GRDDC2022 dataset and the Chinese subset, respectively, outperforming the other detection models. The number of parameters and computations are reduced by 10.3% and 9.9%, respectively, compared to the YOLOv8n model, with an average frame rate of 69 frames per second, offering good real-time performance. In addition, on the CRACK500 dataset, PGS-YOLO improved mAP50 by 2.3%, achieving a better balance between model complexity and detection accuracy.

## 1. Introduction

Currently, the main methods for highway pavement defect detection in China are manual and sensor-based detection. Manual detection can effectively identify pavement defects but has a low detection efficiency, which may lead to traffic congestion or even accidents, and the maintenance costs are high, making it unsuitable for large-scale detection. Sensor-based detection can improve detection efficiency, but specialized road-detection vehicles are expensive. With the rapid development of computer vision technology, the use of deep-learning methods for pavement defect detection has gradually become mainstream. There are two main types of deep-learning-based object detection models: two-stage and single-stage models. Two-stage models obtain candidate regions through convolutional neural networks and then complete classification and localization tasks, such as the fast R-CNN [1], Faster R-CNN [2], and Cascade R-CNN [3] algorithms. Single-stage models use a regression approach, inputting images into convolutional neural networks and directly outputting detection results after processing, such as the YOLO [4] series and SSD [5] algorithms.

## 2. Related Work

Although deep learning methods can solve most of the problems associated with traditional detection methods, their model complexity is high. To address this issue, many researchers have conducted studies in the field of lightweight pavement defect detection. Lin et al. [6] proposed a lightweight pavement defect detection algorithm based on YOLOv5s, which reduces the model complexity by introducing the ghost module from GhostNet and integrating a coordinate attention mechanism to enhance the model’s perception of coefficient features. Although this significantly reduces the model parameters, it also sacrifices the detection accuracy. Diao et al. [7] proposed a lightweight pavement defect detection algorithm, LE-YOLOv5, based on YOLOv5, which reconstructs a feature fusion network by combining GSConv and VCSCSP modules with a parameter-free attention feature fusion module, PAFF. Compared to YOLOv5S, LE-YOLOv5 showed improvements in both detection speed and accuracy. Li et al. [8] proposed a lightweight pavement defect detection model called LHA-Net, which extracts global and local features using the DGFM and HLFM in the feature extraction network. In the feature fusion network, the AWAM mechanism was used for the progressive weighted fusion of multiscale features. Finally, the detection head subnet uses the LDH to separate the tasks and optimize feature extraction. The experimental results show that this model outperforms the original model in terms of both accuracy and complexity, although there is still room for improvement in detection speed.

YOLOv8 is an object detection algorithm launched by Ultralytics, built upon the historical versions of the YOLO series, which introduces new features and improvements, further enhancing performance and flexibility. To address the shortcomings of existing detection models in terms of detection accuracy and model complexity, a lightweight pavement defect detection algorithm, PGS-YOLO, integrating perception enhancement and feature optimization, is proposed based on YOLOv8n. This algorithm reduces the computational load and parameters while improving the average detection accuracy, and achieves a better balance between model complexity and detection accuracy. The work can be summarized as follows:

(1) The original YOLOv8 model has a fixed receptive field, which makes it difficult to adapt to the multi-scale features of pavement defects. Therefore, an efficient Perception-Enhanced Feature Extraction Network (PEFNet) is designed to dynamically adjust the size of the receptive field according to the scale of the input target, enhance the feature extraction capability, and provide higher-quality feature input for the feature fusion module.

(2) Due to the frequent presence of perspective distortion in pavement defect images, traditional FPN+PAN tends to lose details during feature fusion. Therefore, the neck structure is redesigned, and a Generalized Dynamic Sampling Feature Fusion Structure (GDSFPN) is proposed to better retain and transmit features of different resolutions, thereby optimizing the multi-scale feature fusion effect.

(3) Aiming at the problems that the original detection head has multi-scale detection defects, parameter redundancy, and insufficient capture of tiny defects, a Shared Detail-Enhanced Lightweight Detection Head (SDCLD) is designed. It effectively reduces model complexity, accurately extracts pavement defect features, and enhances the accuracy of target detection.

(4) The traditional loss function (CIoU+DFL) has uneven gradient weight distribution for millimeter-level cracks and large localization errors. By replacing the original loss function with Wise-IoUv3, the loss weight of small targets is optimized, the localization ability of the model is improved, and the training effect and detection accuracy of the model are further optimized.

## 3. PGS-YOLO Algorithm

The network structure of the PGS-YOLO algorithm is shown in Figure 1, consisting of the backbone Perception Enhanced Feature Extraction Network (PEFNet), neck feature fusion network (Generalized Dynamic Sampling Feature Pyramid Network, GDSFPN), and head prediction network (Shared Detail-enhanced Convolution Lightweight Detection Head, SDCLD).

### 3.1. Perception Enhanced Feature Extraction Network

Traditional convolution operations typically use fixed parameter settings without considering the differences in pavement defect features at different locations. This leads to low efficiency and poor accuracy when dealing with complex scenes with multiscale features. To address this issue, an efficient perception-enhanced feature extraction network (PEFNet) is proposed, consisting of RFCBAMConv and C2f-RFCBAM modules.

Receptive-Field Attention Convolution (RFA [9]) generates different attention weights for each receptive field by focusing on the spatial features of the receptive field. These weights are multiplied by the convolution kernel parameters to form new convolution parameters, making the convolution operation independent of the shared parameters and offering a new direction for solving the problem of shared convolution kernel parameters. Inspired by this, this idea is applied to improve the Convolutional Block Attention Module (CBAM [10]) attention mechanism. CBAM is a module that combines channel attention and spatial attention; however, it has limitations when handling the issue of shared parameters for large-sized convolution kernels, and its original computation method is relatively complex. To optimize CBAM, two adjustments are made: first, squeeze-and-excitation (SE) attention is used to replace the Channel Attention Module (CAM) in CBAM, effectively reducing the computational cost; second, the method of calculating attention is changed, no longer calculating channel and spatial attention separately, but instead applying weighted attention simultaneously, which allows each channel to obtain a different attention map, enhancing the model’s ability to distinguish features across different channels. On this basis, the idea of focusing on receptive field spatial features through RFA was incorporated, resulting in the design of a new convolution operation, RFCBAMConv. The improved RFCBAMConv module was compared to the CBAM module, as shown in Figure 2.

In RFCBAMConv, a convolution operation with a stride of is used to extract feature information in a manner similar to RFA, generating adaptive parameters based on data distribution and feature composition within the receptive field. This improved the detection accuracy in the initial stage of feature extraction. Simultaneously, multiple dimensions were integrated to weigh the importance of each feature within the receptive field to extract key information. A global pooling strategy was used to deeply mine features within the spatial dimensions of the receptive field, expand the range of feature associations, and enhance the model’s perception of features, significantly improving the ability to extract features at different scales.

RFCBAMConv is used to modify the Bottleneck of the C2f module in the backbone network, leading to the proposal of the C2f-RFCBAM module, whose structure is shown in Figure 1. The redesigned C2f-RFCBAM module can dynamically adjust the receptive field according to the input target scale, accurately extract feature information for targets at different scales, and effectively solve the problem of insensitivity to positional changes caused by parameter sharing in the C2f module. Its refined feature processing ability enhances the detection capability of the model in complex scenarios, thereby improving the accuracy. RFCBAMConv and C2f-RFCBAM are used to form an efficient perception-enhanced feature extraction network (PEFNet), dynamically adjusting the receptive field size based on the input target scale, enhancing feature extraction capabilities, and providing better quality feature input for the feature fusion module.

Through Section 4.4.1 and ablation experiments, it was shown that the feature information processed by PEFNet was effectively enhanced and filtered, allowing better adaptation to the complex forms and scale changes of different pavement defects and providing more discriminative feature representations for subsequent feature fusion.

### 3.2. Generalized Dynamic Sampling Feature Pyramid

Feature pyramids play an important role in object detection models, and their purpose is to aggregate features of different resolutions extracted from the backbone network. The neck of YOLOv8 adopts the FPN [11] + PAN [12] structure, where FPN transmits semantic information through a top-down path, and PAN transmits positional information through a bottom-up path. These two types of features are then fused to enhance feature information at different scales. However, this structure has certain limitations: first, feature maps lose some original feature information after multiple upsampling and downsampling, resulting in poor performance in extracting features for targets with complex shapes; second, it pays less attention to low-level feature maps, easily ignoring detailed information and reducing the accuracy of small target detection.

To address these issues, drawing on the re-parameterized generalized feature pyramid network (RepGFPN) proposed by Xu et al. [13], as shown in Figure 3, we redesign the feature fusion pyramid and propose the Generalized Dynamic Sampling Feature Pyramid (GDSFPN). RepGFPN replaces the original 3 × 3 convolution with CSPNet in feature fusion blocks, upgrades CSPNet by combining the re-parameterization mechanism with efficient layer aggregation network connections, sets different channels for feature maps of different scales, and removes additional upsampling operations to improve real-time performance.However, this structure has insufficient upsampling operations, making it difficult to restore the resolution of feature maps, which leads to the loss of feature information and affects the detection of small targets and targets with complex shapes.

In view of this, the dynamic upsampling module DySample is introduced to optimize RepGFPN, thus constructing GDSFPN. The schematic diagram of the GDSFPN network structure is shown in Figure 4, which retains the CSPStage and core structures of re-parameterized convolution from RepGFPN.

The combination mechanism of DySample and RepGFPN is as follows: In the original structure of RepGFPN, when features are transmitted from high-level to low-level layers, scale matching is achieved through static upsampling with fixed multiples, and this process tends to lose detailed information.GDSFPN replaces all these static upsampling layers with DySample, which adaptively adjusts sampling positions and weights according to the semantic distribution of input features through a dynamic sampling point generation mechanism, while improving resolution and retaining key features.For example, when processing small-scale crack features, DySample will focus on sampling points in the edge regions of cracks to enhance detail transmission; for large-scale pothole features, it maintains overall morphological information through a global sampling strategy.This modification retains the advantages of RepGFPN, while improving the recovery effect of fine-grained features, especially for small defects such as longitudinal cracks (D00) and defects with complex shapes such as reticular cracks (D20).

Compared with traditional kernel-based upsampling techniques, DySample achieves efficient upsampling through a unique point sampling method, reducing computational load and resource requirements. It utilizes dynamic convolution and point sampling strategies, without the need for computationally intensive processes or additional sub-networks, and exhibits strong flexibility and adaptability. It generates rich mappings from the input low-resolution feature maps through convolution or feature extraction, dynamically generates sampling points to meet upsampling needs, and effectively improves image resolution. Figure 5 shows the dynamic upsampling technology and the design of its related modules.

Through Section 4.4.2 and ablation experiments, it is shown that the dynamic sampling mechanism of the GDSFPN makes full use of the multi-scale features provided by PEFNet, effectively compensating for the loss of feature information in traditional feature pyramids during upsampling and downsampling. This ensures that feature information from different layers can fully participate in the subsequent detection process, enhancing the ability to detect small and complex-shaped objects and providing more comprehensive feature input for the detection head.

### 3.3. Shared Detail-Enhanced Lightweight Detection Head

The original detection head of YOLOv8 uses a single-scale structure, with each detection head receiving its own feature-map input. This structure leads to poor detection accuracy for multiscale targets, as there is a lack of information exchange between detection heads at different levels, resulting in the loss of some detailed information. In addition, targets of different sizes are processed through independent convolutional layers, each with its own parameters, which significantly increases the number of parameters and computations.

To address the defects in the original detection head, a Shared Detail-Enhanced Lightweight Detection Head (SDCLD) was redesigned. It uses Detail-Enhanced Convolution (DEConv) to modify the detection head. Compared to ordinary convolution, DEConv is better at highlighting high-frequency information and has a stronger ability to capture the detailed features of small targets. Its structure is shown in Figure 6.

DEConv consists of five parallel convolution layers: standard convolution (VC), central difference convolution (CDC), angle difference convolution (ADC), horizontal difference convolution (HDC), and vertical difference convolution (VDC). These convolutions capture image features from different angles and dimensions while utilizing reparameterisation techniques. Given the input feature $F_{\mathrm{in}}$, DEConv can output $F_{\mathrm{out}}$ with the same computational cost and inference time as regular convolution layers, as defined by the following formula: (1)$$F_{\mathrm{out}}=\mathrm{DEConv}\left(F_{\mathrm{in}}\right)=\sum_{i=1}^{5}F_{\mathrm{in}}*K_{i}=F_{\mathrm{in}}*\left(\sum_{i=1}^{5}K_{i}\right)=F_{\mathrm{in}}*K_{\mathrm{cvt}}$$
where DEConv() represents the DEConv operation, $K_{i}=1,5$ represents the kernels of VC, CDC, ADC, HDC, and VDC, ∗ denotes the convolution operation and $K_{\mathrm{cvt}}$ represents the transformed kernel, which combines the parallel convolutions.

Additionally, Group Normalization (GN) and Shared Convolution (SConv) structures were used to improve the localization and classification performance of the detection head while significantly reducing the number of parameters. Finally, the channels were adjusted and the features were scaled to accurately predict the size and category of the bounding boxes. The structure of the detection head is shown in Figure 7.

Through Section 4.4.3, it can be seen that using DEConv to improve the detection head effectively enhances the detection capability for pixel-level small targets. At the same time, Group Normalization and Shared Convolution structures significantly reduce the number of parameters in the detection head, thereby achieving the goal of lightweight design.

### 3.4. WIoU Loss Function

The loss function of YOLOv8 includes classification loss (VFL) and regression loss (CIOU + DFL). When detecting small targets, it is easy to ignore the feature and localization differences between targets of different sizes, which leads to missed and false detections. To address this, the WIoU [14] was used to replace the CIoU. The WIoU uses a non-monotonic focusing mechanism and improves the anchor-box quality evaluation process by generating dynamic gradient gain factors. This allows for a more effective handling of samples with different qualities, thereby improving the overall performance of the model. The schematic diagram of the WIoU bounding box loss is as shown in Figure 8, and its formula is as follows:(2)$$L_{WIoUv1}=R_{WIoU}L_{IoU}$$(3)$$R_{WIoU}=\exp\left(\frac{{\left(x-x_{\mathrm{gt}}\right)}^{2}+{\left(y-y_{\mathrm{gt}}\right)}^{2}}{{\left(W_{g}^{2}+H_{g}^{2}\right)}^{*}}\right)$$(4)$$L_{IoU}=1-IoU$$(5)$$IoU=\frac{A\cap B}{A\cup B}$$

In the formula: (x,y) are the coordinates of the center of the anchor box; $\left(x_{\mathrm{gt}},y_{\mathrm{gt}}\right)$ are the coordinates of the target box; $W_{g}$ and $H_{g}$ are the dimensions of the minimum enclosing box; ∗ denotes the separation of $W_{g}$ and $H_{g}$ from the computation graph; $L_{IoU}$ is used to reduce the $R_{WIoU}$ of high-quality anchor boxes, and $R_{WIoU}$ is used to amplify the $L_{IoU}$ of regular-quality anchor boxes.

WIoUv3 introduces a nonmonotonic focusing coefficient based on WIoUv1, allowing WIoUv3 to dynamically adjust the gradient gain during training. It assigns smaller gradient gains to anchor boxes that match the target box well (low outlier degree), thus making the bounding box regression focus more on anchor boxes of an average quality. Conversely, larger gradient gains were assigned to anchor boxes that poorly matched the target box (high outlier degree), preventing low-quality anchor boxes from causing overly large negative impacts.(6)$$L_{WIoUv3}=r\cdot L_{WIoUv1},\;r=\frac{\beta }{\delta \alpha^{\beta -\delta }}$$(7)$$\beta =\frac{L_{IoU}^{*}}{L_{IoU}}\in \left[0,+\infty \right)$$

In the formula, α and β are hyperparameters, with optimal abnormal anchor box suppression effects when set to 1.9 and 3.0 and β represents the outlier degree, with smaller outlier degrees indicating higher anchor box quality. $L_{\mathrm{IoU}}^{*}$ is the monotonic focal coefficient, which can be dynamically adjusted based on the detection target. ${\bar{L}}_{\mathrm{IoU}}$ is a normalization factor, representing the moving average value of momentum, and can adjust the maximum gradient gain. Because $L_{\mathrm{IoU}}$ and the anchor box quality standard are dynamic, WIoUv3 can dynamically allocate gradient boosting to address the slow convergence problem during the later stages of training. Through an experimental comparison, introducing WIoUv3 into YOLOv8 to replace CIoU shows significant improvement, as the model focuses more on normal-quality anchor boxes, improves target localization ability, dynamically optimizes the loss weight for small targets, and enhances the model’s detection performance.

## 4. Experiments and Analysis

### 4.1. Dataset

The experimental dataset was obtained from the open-source dataset provided by the GRDDC2022 [15] Global Road Damage Detection Challenge, which includes road images from multiple countries. A subset of 4373 images from the Chinese portion of the dataset was selected for the experiment. Among them, 2396 images were captured by drones and 1977 images were captured by vehicle-mounted cameras. There are four types of pavement defects: longitudinal cracks (D00), transverse cracks (D10), Networked cracks (D20), and potholes (D40), as shown in Figure 9. The dataset was randomly split into training, validation, and test sets at an 8:1:1 ratio.

The distributions of the different pavement defect types in the dataset are presented in Table 1.

### 4.2. Experimental Platform and Hyperparameter Settings

All experiments were conducted under the same training hyperparameters and experimental environment to ensure objectivity and fairness of the experimental results. The specific experimental environment configurations and training hyperparameters are listed in Table 2 and Table 3, respectively.

In addition, the SGD optimizer is selected. On the one hand, because the YOLOv8n baseline model and comparative algorithms (YOLOv5s+Ghost+CA, LE-YOLOv5) often adopt this optimizer, it can ensure the fairness of experimental comparison. On the other hand, on the small-scale dataset used in this study, SGD is less prone to overfitting than adaptive optimizers, which is beneficial to the model’s generalization learning of fine-grained defect features.

### 4.3. Evaluation Metrics

To objectively evaluate the model’s detection performance, the following evaluation metrics were used: Precision, Recall, mean Average Precision (mAP), parameters, Giga Floating-Point Operations Per Second (GFLOPS), and Frames Per Second (FPS).

Precision measures the proportion of true positive samples among all samples predicted as positive by the model; recall measures the proportion of true positive samples among all actual positive samples correctly predicted by the model; Precision and Recall have a trade-off, where increasing precision may reduce recall, which is typically quantified by the F1-score; and mAP is used to measure the model’s accuracy. The calculation formulas for the evaluation metrics are as follows.(8)$$\mathrm{Precision}=\frac{TP}{TP+FP}$$(9)$$\mathrm{Recall}=\frac{TP}{TP+FN}$$(10)$$F1\text{-}\mathrm{score}=\frac{2\cdot \mathrm{Precision}\cdot \mathrm{Recall}}{\mathrm{Precision}+\mathrm{Recall}}$$(11)$$mAP=\frac{\sum P_{A}}{N}$$

GFLOPS was used to estimate the execution time of the model, and the parameters measured the size and complexity of the model.

### 4.4. Experimental Results and Analysis

#### 4.4.1. Experimental Comparison of Attention Modules and Feature Extraction Networks

To verify the performance advantage of the constructed PEFNet, a comparison was performed with other mainstream attention modules on the GRDDC2022 Chinese subset under the same experimental conditions. The bottlenecks of the Conv and C2f modules were replaced with improvements at the same location. The experimental results are listed in Table 4.

In the table, YOLOv8n is the baseline model, and “+” indicates that the corresponding module improvements were made based on YOLOv8n. The experimental results show that the computational load with the PEFNet backbone network increases by only 0.7 GFLOPS, and mAP50 improves by 1.9% compared with the baseline. The results demonstrate the effectiveness of using RFCBAMConv to improve the convolution and bottleneck in the original backbone and C2f modules.

#### 4.4.2. Verification of GDSFPN Feature Fusion Structure

To validate the effectiveness of the GDSFPN, a performance comparison experiment was conducted between the original feature fusion structure PAFPN in YOLOv8 and the proposed GDSFPN. The experimental results are listed in Table 5. The experimental results showed that when YOLOv8n used the GDSFPN structure for feature fusion, the parameter and computational loads increased by only 0.25 M and 0.2 G, and mAP50 increases by 1.7%. This indicates that the GDSFPN structure achieves better feature fusion and significantly improves average detection accuracy.

To intuitively demonstrate the improvement of the GDSFPN structure in feature fusion performance, Grad-CAM [16] technology is used for visualization analysis, generating detection heatmaps of the feature fusion structures of GDSFPN and PAFPN, as shown in Figure 10. Visualization results for defects show that: for tiny longitudinal cracks, the heatmap of GDSFPN has higher attention weights in the edge regions of the cracks, indicating that it more accurately focuses on defect details; for reticular cracks with complex shapes, the attention distribution of GDSFPN’s heatmap is more concentrated and less affected by background interference.

It can be clearly observed from the heatmaps that the PAFPN structure is prone to interference from the background (such as vehicles, shadows), and its attention tends to focus on areas outside the target, which affects detection accuracy. In contrast, the proportion of attention of the GDSFPN structure to pavement defect areas has significantly increased, and its attention to pothole areas is significantly higher than that of PAFPN. This further confirms that GDSFPN has optimized the multi-scale feature fusion effect, enhanced the model’s targeted attention to defect areas, and provided a clearer interpretability basis for model decision-making.

#### 4.4.3. Verification of SDCLD Lightweight Detection Head Effectiveness

The computation load of the YOLOv8 detection head accounts for 40% of the total; therefore, lightweight processing of the YOLOv8 model’s detection head is key to reducing the overall model size. To verify the lightweight advantage of the designed SDCLD detection head, it was compared experimentally with the original detection head and the RSCD lightweight detection head proposed by Cao et al. [17]. The experimental results are listed in Table 6.

The experimental results show that compared to the baseline model, the SDCLD detection head improves the detection accuracy for the D00 and D10 classes, which mostly consist of small cracks, by 2.7% and 2.1%, respectively. This proves that DEConv effectively enhances the ability of the model to capture small target details, with mAP50 improving by 1.5%, and the number of parameters and computation were significantly reduced by 21.6% and 19.8%, respectively. Compared with the RSCD detection head, SDCLD exhibits significant advantages in all aspects.

#### 4.4.4. Validation of WIoU Effectiveness

To verify the effectiveness of Wise-IoU compared with traditional loss functions, comparative experiments are conducted on the Chinese subset of GRDDC2022, using CIoU, DIoU, and WIoUv3 as loss functions respectively, with the remaining experimental conditions kept consistent. The experimental results are shown in Table 7.

The experimental results show that for small targets such as longitudinal cracks (D00) and transverse cracks (D10), the AP% of WIoU reaches 86.2 and 84.5, which is an increase of 2.1% and 2.6% compared with CIoU, and 1.8% and 1.3% compared with DIoU. For large targets including reticular cracks (D20) and potholes (D40), the AP% of WIoU reaches 76.1% and 75.3%, showing an increase of 1.7% and 2.6% compared with CIoU, and 1.2% and 1.5% compared with DIoU.

Wise-IoUv3 dynamically adjusts gradient gain through a non-monotonic focusing mechanism, assigning larger gradient weights to anchor boxes with low matching degrees and reducing gradient interference for anchor boxes with high matching degrees. This enables the model to more accurately optimize bounding box regression during training.In contrast, although CIoU takes into account the overlap degree of bounding boxes, center point distance, and aspect ratio, it has insufficient gradient optimization for low-quality anchor boxes; DIoU focuses on center point distance optimization but neglects the dynamic adjustment of anchor box quality.The experimental results show that in the detection of different types of defects, the mAP50 of Wise-IoUv3 is consistently higher than that of CIoU (1.5%) and DIoU (1.1%), demonstrating its stability and superiority in improving detection accuracy and further optimizing the training effect of the model.

#### 4.4.5. Ablation Experiment

To verify the effectiveness of the proposed improvement modules, ablation experiments were designed to compare the impacts of different module combinations on the performance of the pavement defect detection algorithm. YOLOv8n was used as the baseline model for the ablation experiment, and the validation was performed on the validation set under the same experimental conditions. The results are presented in Table 8.

As shown in the table, compared to the baseline model, when using the Perception Enhanced Feature Extraction Network (PEFNet) as the model’s feature extraction module, the detection accuracy for complex and large-scale defects in the D20 and D40 categories improves by 2.9% and 3.7%, respectively. The number of parameters and computations increased slightly by 4% and 8%, respectively. When the redesigned GDSFPN is used as the feature fusion module, the detection performance for D20 and D40 defects is also improved, with the detection accuracy increasing by 3.7% and 2.9%, and the parameters and computation increased by only 8% and 2%, respectively. Using SDCLD improves the detection accuracy for small cracks in the D00 and D10 categories by 2.7% and 2.1%, respectively, while reducing the number of parameters and computations by 21.6% and 19.8%, respectively. When using WIoUv3 to replace the original loss function, the detection accuracy for D00 and D10 defects, which mainly consist of small targets, improved by 2.4% and 1.5%, respectively. For D20 defects, which have large-scale variations, the detection accuracy improves by 0.7%, and mAP50 increases by 0.9%, proving that WIoUv3 can dynamically optimize small target loss weights and enhance the localization capability. When PEFNet and GDSFPN are used together, the detection accuracy for D20 and D40 defects, which are complex in shape and have large-scale variations, improves by 2.9% and 2.3%, respectively, and mAP50 improves by 1.9%. The parameters and computation increase by only 11% and 8%, respectively, proving that the feature information processed by PEFNet is better fused by the GDSFPN without adding extra computational burden. Finally, when PEFNet, GDSFPN, and the lightweight detection head SDCLD were used together, the detection capability for various defects improved, with the mAP50 increasing by 2.4%, and the parameters and computation were reduced by 10.3% and 9.9%, respectively. The improved PGS-YOLO model, with WIoUv3, achieved the highest average detection accuracy of 82.3%, with mAP50 improving by 2.9% compared to the baseline model, and FPS reaching 69 frames per second, meeting real-time detection requirements. It balances model complexity and detection accuracy, while further improving accuracy and reducing parameters and computation.

#### 4.4.6. Comparison of Different Algorithms

To comprehensively evaluate the performance of the PGS-YOLO model, it was compared with the current mainstream object detection algorithms and lightweight algorithms. The comparison of mAP_0.5 and mAP_0.5:0.95 curves of different models is shown in Figure 11. Additionally, lightweight pavement defect detection algorithms, YOLOv5s+Ghost+CA and LE-YOLOv5, were also compared. The experiments were conducted under the same conditions, and all models converged. The results are presented in Table 9.

The experimental results show that, in terms of accuracy, PGS-YOLO performs best on evaluation metrics, such as Precision, Recall, mAP50, and F1-score, with mAP50 being the highest at 82.3%. In terms of model complexity, PGS-YOLO also performs excellently, with parameters and computation of 2.7M and 7.3G, respectively, which is only higher than those of YOLOv5n, YOLOv9t, YOLOv10n, and YOLOv11n. However, the detection accuracy improved by 4.7, 4.5, 4.6, and 3.7%, respectively. Compared to YOLOv8n, the number of parameters and computations are reduced by 10.3% and 9.9%, respectively.

Compared to YOLOv5s+Ghost+CA and LE-YOLOv5, which are also used in pavement defect detection, the mAP50 of PGS-YOLO was 1.3% and 0.5% higher, respectively. The number of parameters and computation is reduced by 27.6% and 21.5%, respectively, and the detection frame rate is improved by 20.3% and 11.6%, respectively, reaching 69 frames per second. This indicates that PGS-YOLO achieves a better balance between real-time performance and accuracy through a reasonable module design, while pursuing higher precision. The experiments prove that PGS-YOLO has advantages in terms of model complexity, detection accuracy, and speed, and achieves a better balance between detection accuracy and model complexity.

#### 4.4.7. Comparison Across Different Datasets

To verify the cross-dataset generalization ability of the algorithm and its applicability in pavement defect detection, comparative experiments were conducted on the complete GRDDC2022 [15] dataset and the CRACK500 dataset proposed by Yang et al. [27] with current mainstream object detection algorithms.

The GRDDC2022 dataset includes 47,420 road images from Japan, India, the Czech Republic, Norway, the United States, and China, covering four categories: longitudinal cracks (D00), transverse cracks (D10), network cracks (D20), and potholes (D40). As a large number of images in it did not have required annotations, 23,767 images were screened for comparative experiments. The dataset was divided into training, validation, and test sets using a random division method at a ratio of 8:1:1, and the experimental results are shown in Table 10.

The experimental results show that PGS-YOLO outperforms other detection models in evaluation metrics such as P%, R%, mAP50, and F1 score.In terms of detection speed, its FPS reaches 64 frames per second, which is lower than that of YOLOv10n (85FPS) but higher than that of RT-DETR (36FPS) and YOLOv11 (44FPS). Considering the large scale and diverse scenarios of the complete GRDDC2022 dataset, the FPS of PGS-YOLO fully meets the real-time detection requirements in practical engineering.Meanwhile, compared with YOLOv8n, its GFLOPs are reduced by 20%, and mAP50 is increased by 2.8%, reaching 58.0%, the highest among all compared models. This indicates that PGS-YOLO maintains high efficiency while improving accuracy. This result demonstrates that PGS-YOLO has good generalization ability and excellent applicability in pavement defect detection.

The initial Crack500 dataset includes 500 road crack images, with each image generally having a resolution of 1440 × 2560 or 2560 × 1440. Due to the limited number of images, large size of each image, and limited computing resources, the original images are cropped into 16 non-overlapping sub-regions according to a 4 × 4 grid. Only sub-regions with more than 1000 crack pixels are retained, while regions with no cracks or too few cracks are discarded. Finally, rotation or cropping is used to ensure the final sub-regions have a resolution of 640 × 360. Since the original labels of this dataset are not suitable for pavement defect detection, the LabelImg tool is used for manual annotation. Following the annotation specifications of the GRDDC2022 dataset, defects are classified into four categories: D00 (longitudinal cracks), D10 (transverse cracks), D20 (reticular cracks), and D40 (potholes). Finally, 3368 road crack images meeting the annotation requirements for object detection are obtained. The dataset is randomly divided into a training set, validation set, and test set in an 8:1:1 ratio, and the experimental results are shown in Table 11.

The experimental results show that, compared with current mainstream object detection algorithms, PGS-YOLO also performs excellently on the processed CRACK500 dataset, with its P%, R%, and mAP50 reaching 68.0%, 38.8%, and 45.2% respectively, all being the highest values.Compared with the YOLOv8n model, the detection accuracy is improved by 2.3%, and the FPS reaches 72 frames per second, which is higher than that of RT-DETR (38FPS) and YOLOv11 (58FPS), and close to that of YOLOv10n (86FPS). Its GFLOPs remain at a low level, ensuring efficient computation in the fine-grained crack detection required by CRACK500. This experimental result demonstrates that PGS-YOLO has good cross-dataset generalization ability and achieves a balance between detection accuracy and efficiency.

## 5. Pavement Defect Detection Visualization and Analysis

A comparison of the detection performance with current mainstream detection models is conducted to objectively evaluate the performance of the PGS-YOLO algorithm in real-world scenarios. Considering the impact of different scenarios on the detection performance of the model, one drone-captured image and one vehicle-mounted camera-captured image were randomly selected from the Chinese subset of the GRDDC2022 dataset as experimental samples.

### 5.1. Drone-Captured Road Condition Images

As shown in Figure 12, when detecting drone-captured images, YOLOv5n, YOLOv8n, YOLOv10n, YOLOv11n, and RT-DETR all showed cases of missed detection. YOLOv5n, YOLOv10n, and YOLOv11n also suffer from multiple detections of the same target, resulting in overlapping detection boxes. In contrast, PGS-YOLO performed excellently with almost no false positives or missed detections, and the confidence in the detection results was significantly improved.

### 5.2. Vehicle-Captured Road Condition Images

As shown in Figure 13, when detecting vehicle-mounted camera images, YOLOv5n, YOLOv8n, YOLOv10n, and YOLOv11n all had missed detection issues, with YOLOv10n exhibiting the most prominent missed detection problem, significantly missing more targets than the other detection models. YOLOv5n also showed false-positive results. Meanwhile, the YOLOv10n, YOLOv11n, and RT-DETR models exhibited overlapping detection boxes, severely affecting the accuracy and reliability of the detection results.

Combining the results of both experiments, it can be observed that the PGS-YOLO model performs excellently in different scenarios, achieving precise detection of challenging fine cracks and small target defects with no missed or false detections. Additionally, there are no issues with overlapping detection boxes, and the accuracy of the generated detection boxes in terms of location and size is significantly better than that of the other detection models. This set of experiments fully verifies the superior performance of the PGS-YOLO model in various detection scenarios and provides an effective guarantee for multiscenario detection in practical pavement defect detection tasks.

## 6. Conclusions

To address the problem of a large computational load and the difficulty of balancing model complexity with detection accuracy in current pavement defect detection models, an improved algorithm, PGS-YOLO, is proposed. The algorithm first designs a receptive field attention module, C2f-RFCBAM, which replaces the C2f module and reconstructs the entire backbone network to form the perception-enhanced feature extraction network (PEFNet), enhancing the network’s ability to capture details and distinguish features. Next, dynamic upsampling is introduced into the feature fusion network, and a new Generalized Dynamic Sampling Feature Pyramid is proposed to optimize the multiscale feature fusion effect. In addition, the detection head structure was redesigned to significantly reduce the number of parameters while improving the localization and classification performance. Finally, WIoUv3 was used to optimize the small target loss weights and improve the training results. Experimental results on the public GRDDC2022 China subset show that the PGS-YOLO algorithm achieves Precision, Recall, and mAP50 values of 81.8%, 74.5%, and 82.3%, respectively, which are improvements of 0.7%, 1.7%, and 2.9%, respectively, compared to YOLOv8n. The number of parameters decreases from 3.01 M to 2.7 M, a reduction of 10.3%, while the computational load is reduced by 9.9%, and the frame rate reaches 69 frames per second, fully meeting real-time detection requirements. At the same time, on the full GRDDC2022 dataset, mAP50 improved by 2.8% compared with the baseline model, showing good results. Furthermore, on the CRACK500 dataset, mAP50 of PGS-YOLO increased by 2.3%. In conclusion, the proposed algorithm achieves a better balance between model complexity and detection accuracy, with a strong practical value.

Although the improved model has significantly enhanced detection accuracy on public road defect datasets, in practical applications, due to the diversity of road types (such as expressways, urban roads, rural roads, etc.), the complex and variable forms of defects (such as mixed interference between cracks and water stains, oil stains), and frequent exposure to extreme scenarios like rainy and foggy weather and vehicle occlusion, the detection robustness of the improved model still needs further verification.Future research will focus on exploring semi-supervised learning and self-supervised learning strategies to improve the model’s adaptability to complex scenarios. For example, a self-supervised learning framework can be used to automatically generate simulated data labels under harsh conditions such as rain, fog, and occlusion, and combine a small number of real labeled samples to build a semi-supervised training set, thereby alleviating the problem of data scarcity in extreme scenarios.Meanwhile, research on pre-trained networks based on contrastive learning will enable them to learn general feature representations from unlabeled road images, enhancing noise resistance to interferences such as oil stain occlusion and rain/fog blur.Through these advanced learning strategies, it is expected to further optimize the model’s detection performance in complex environments and improve its generalization ability and robustness in actual road scenarios.

## Figures and Tables

**Figure 1 sensors-25-04443-f001:**
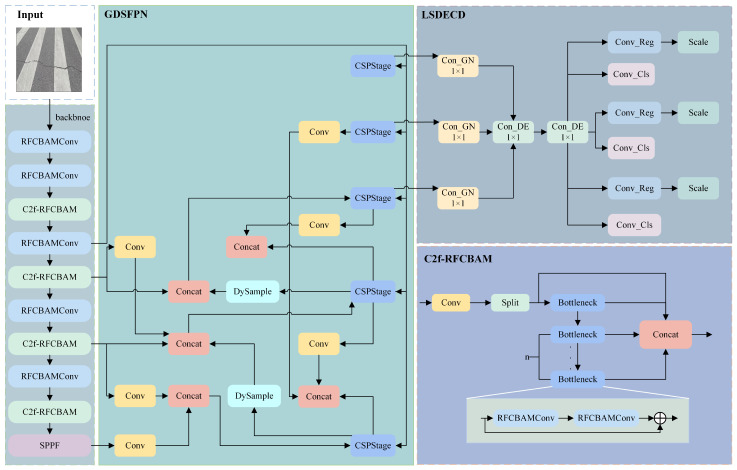
Improved network structure.

**Figure 2 sensors-25-04443-f002:**
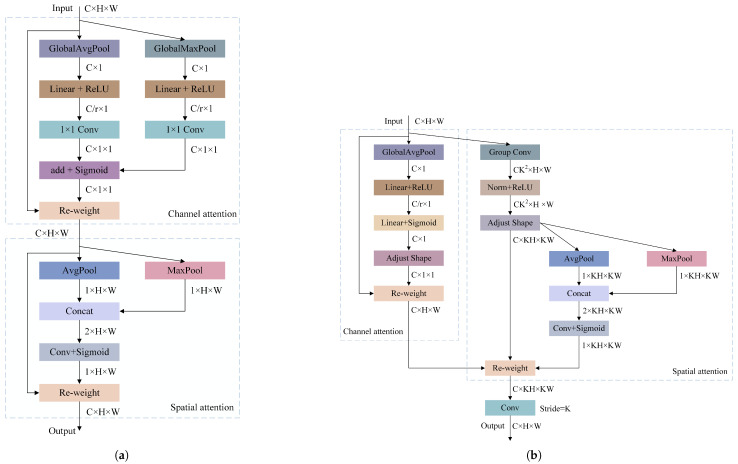
Comparison of CBAM and RFCBAMConv structures: (**a**) CBAM structure diagram. (**b**) RFCBAMConv structure diagram.

**Figure 3 sensors-25-04443-f003:**
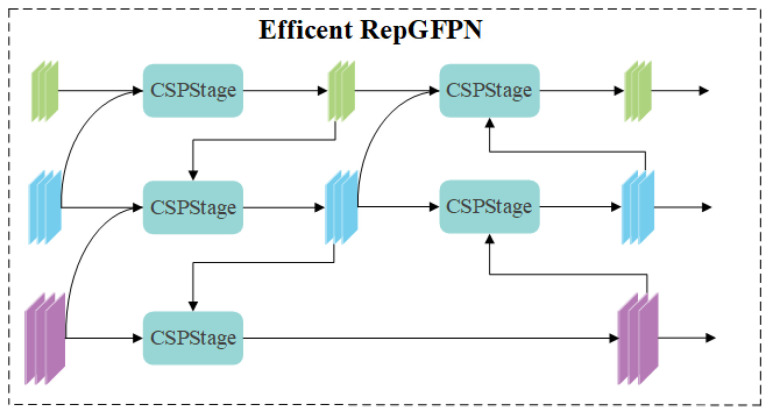
RepGFPN structure.

**Figure 4 sensors-25-04443-f004:**
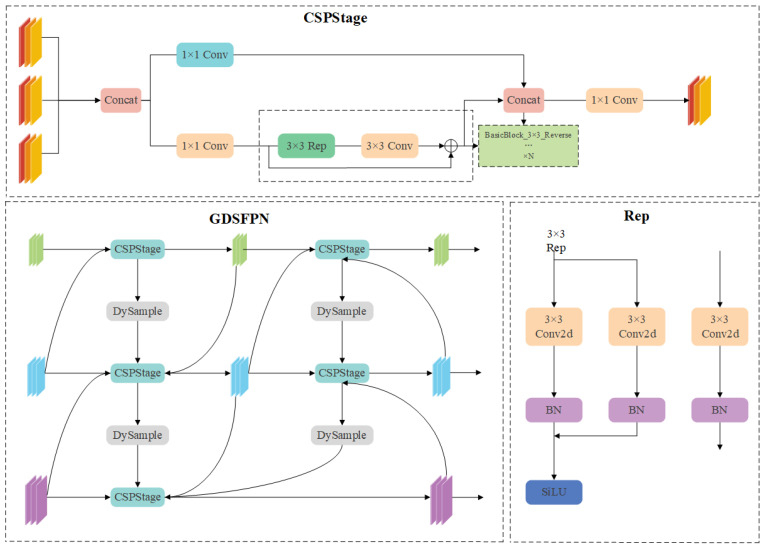
GDSFPN structure.

**Figure 5 sensors-25-04443-f005:**
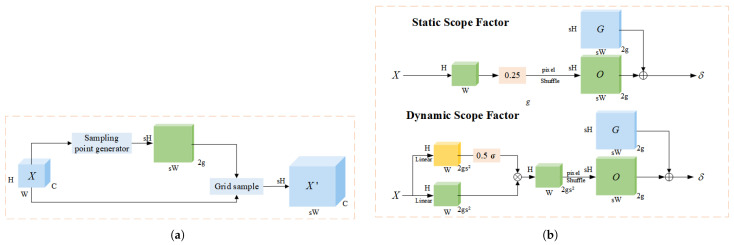
Comparison of CBAM and RFCBAMConv structures: (**a**) CBAM structure diagram. (**b**) RFCBAMConv structure diagram.

**Figure 6 sensors-25-04443-f006:**
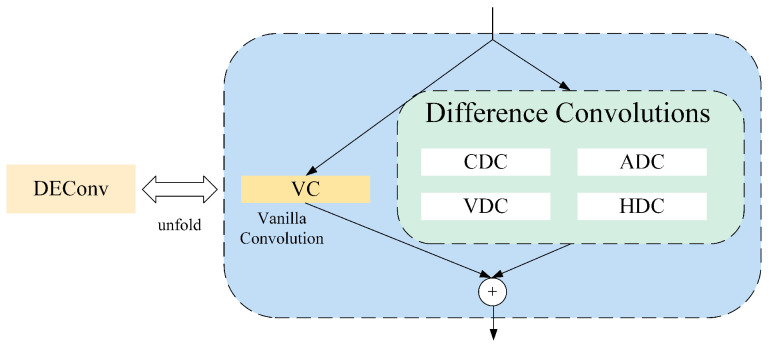
DEConv structure.

**Figure 7 sensors-25-04443-f007:**
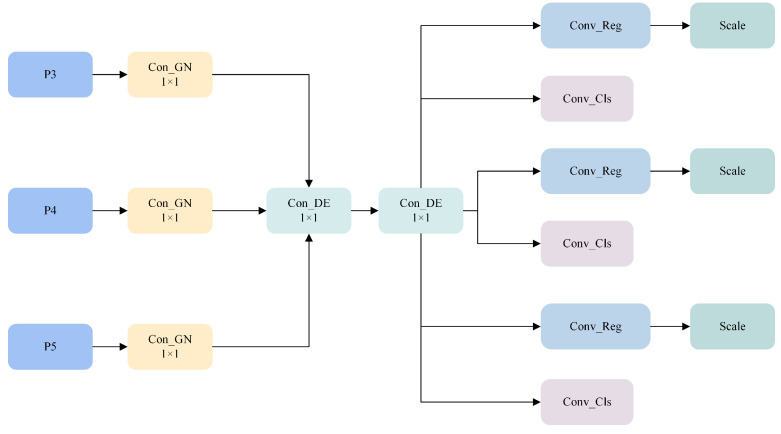
SDCLD structure.

**Figure 8 sensors-25-04443-f008:**
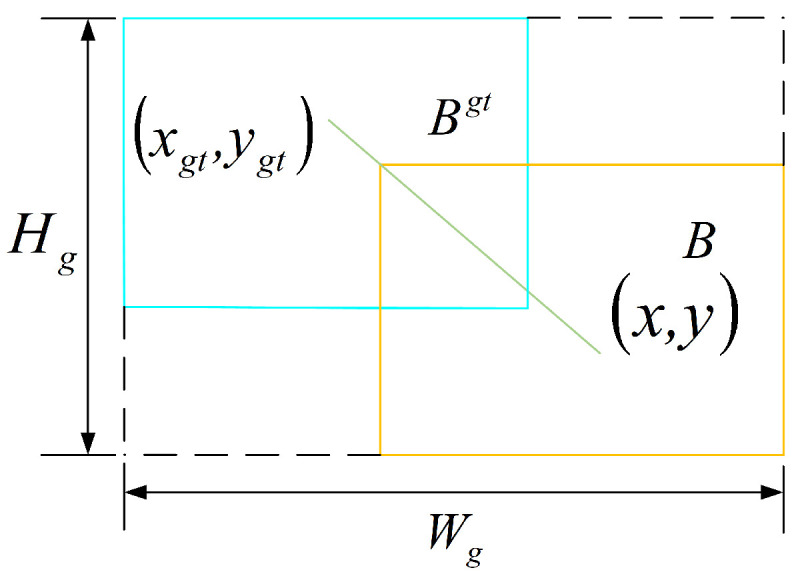
Boundary box loss diagram.

**Figure 9 sensors-25-04443-f009:**
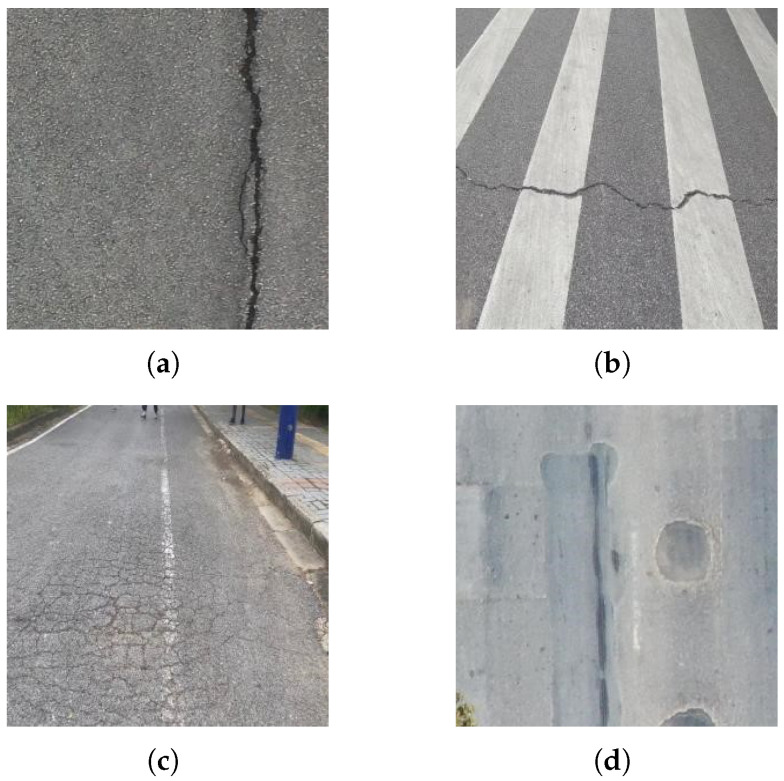
Examples of different pavement defects: (**a**) Longitudinal cracks. (**b**) Lateral cracks. (**c**) Networked cracks. (**d**) Pits.

**Figure 10 sensors-25-04443-f010:**
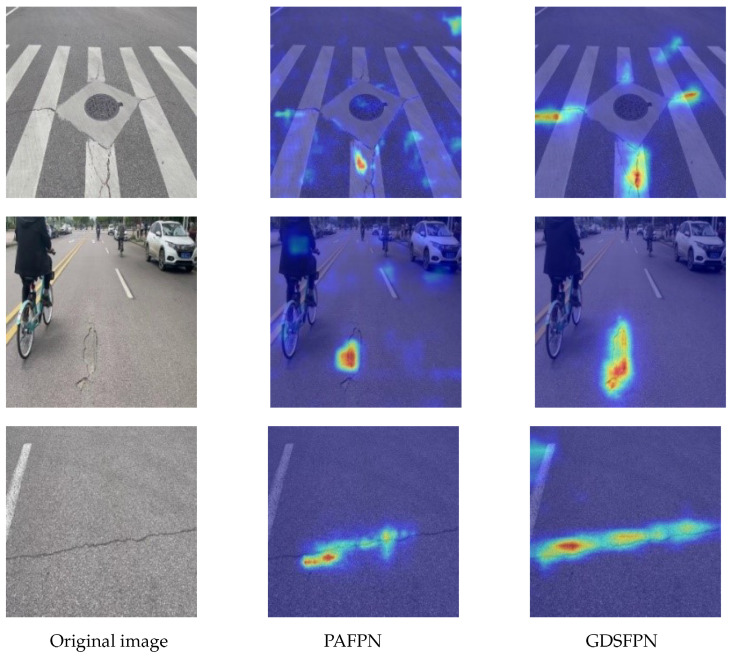
Heatmap comparison of PAFPN and GDSFPN.

**Figure 11 sensors-25-04443-f011:**
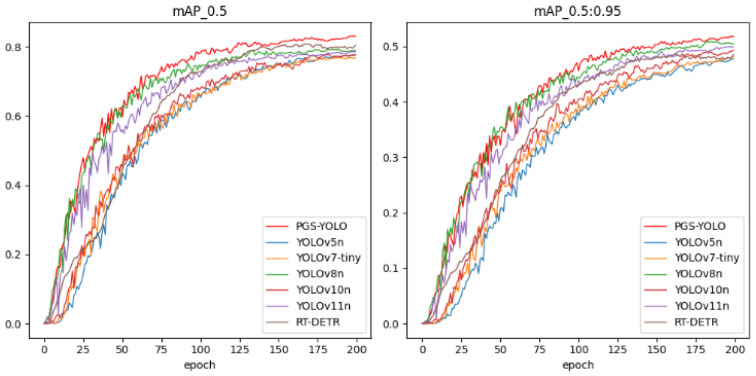
Comparison of mAP_0.5 and mAP_0.5:0.95 curves for different models.

**Figure 12 sensors-25-04443-f012:**
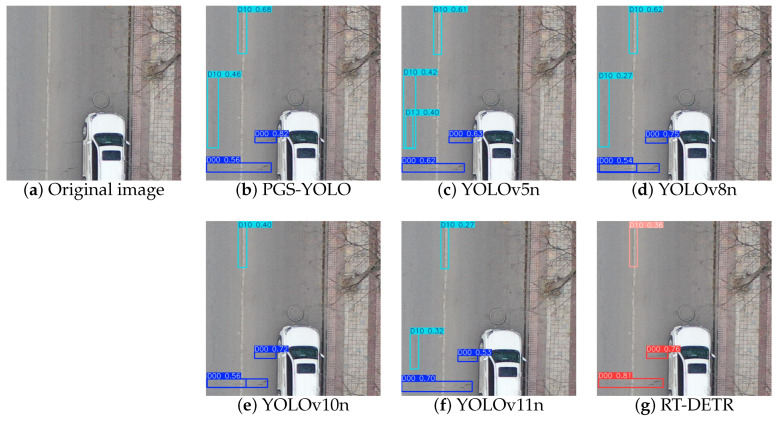
Visualization comparison of the different models: (**a**) Original image:D00 and D10. (**b**) PGS-YOLO. (**c**) YOLOv5n. (**d**) YOLOv8n. (**e**) YOLOv10n. (**f**) YOLOv11n. (**g**) RT-DETR.

**Figure 13 sensors-25-04443-f013:**
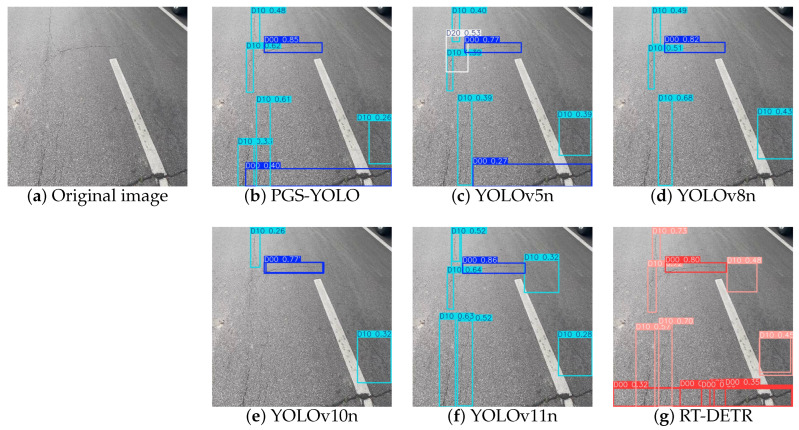
Visualization comparison of the different models: (**a**) Original image:D00 and D10. (**b**) PGS-YOLO. (**c**) YOLOv5n. (**d**) YOLOv8n. (**e**) YOLOv10n. (**f**) YOLOv11n. (**g**) RT-DETR.

**Table 1 sensors-25-04443-t001:** Types of Defects and Quantities in the Dataset.

Defect Type	Label	Quantity
Longitudinal Cracks	D00	2359
Lateral Cracks	D10	4104
Networked Cracks	D20	934
Pits	D40	321

**Table 2 sensors-25-04443-t002:** Experimental Setup.

Name	Parameter
Operating System	Windows 11
CPU	Intel (R) Core (TM) i9-13900HX, Santa Clara, CA, USA
GPU	NVIDIA GeForce RTX 4060, 8 GB, Santa Clara, CA, USA
Memory	16 GB
Programming Language	Python 3.9
Framework	Torch 2.2.2 + cuda121

**Table 3 sensors-25-04443-t003:** Training Hyperparameters.

Parameter	Value
Image Size	640 × 640
Workers	8
Batch Size	16
Epoch	200

**Table 4 sensors-25-04443-t004:** Validation of PEFNet Effectiveness.

Group	Models	mAP50%	Params (M)	FLOPS (G)
Baseline	YOLOv8n	79.4	3.01	8.1
1	+CA	79.9	7.42	25.1
2	+CBAM	80.2	5.78	21.3
3	+RFCAConv	80.3	3.10	8.5
4	+RFCBAMConv	80.7	3.07	8.4
5	+C2f-RFCA	80.1	3.05	8.5
6	+C2f-RFCBAM	80.9	3.05	8.4

**Table 5 sensors-25-04443-t005:** Performance Comparison between GDSFPN and PAFPN.

Feature Fusion Structure	mAP50%	Params (M)	FLOPS (G)
PAFPN	79.4	3.01	8.1
GDSFPN	81.1	3.26	8.3

**Table 6 sensors-25-04443-t006:** Comparison of Different Lightweight Detection Heads.

Detection Head	AP%				mAP50%	Params	FLOPS
	D00	D10	D20	D40			
Baseline	83.8	83.0	75.4	75.3	79.4	3.01 M	8.1
RSCD	85.0	80.1	70.4	71.7	76.8	2.36 M	6.5
SDCLD	86.5	85.1	75.9	76.1	80.9	2.36 M	6.5

**Table 7 sensors-25-04443-t007:** Loss Function Comparison.

Model	AP%				mAP_0.5	Params	GFLOPS	FPS
	D00	D10	D20	D40				
YOLOv8 (Baseline)	83.8	83.0	75.4	75.3	79.4	3.01 M	8.1	92
YOLOv8+CIoU	84.1	81.9	74.4	72.7	78.3	3.01 M	8.1	83
YOLOv8+DIoU	84.4	83.2	74.9	73.8	79.1	3.01 M	8.1	75
YOLOv8+WIoUv3	86.2	84.5	76.1	75.3	80.5	3.01 M	8.1	90

**Table 8 sensors-25-04443-t008:** Ablation Experiment.

Model	AP%				mAP_0.5	Params	GFLOPS	FPS
	D00	D10	D20	D40				
YOLOv8	83.8	83.0	75.4	75.3	79.4	3.01 M	8.1	92
+SDCLD	86.5	85.1	75.9	76.1	80.9	2.36 M	6.5	110
+WIoUv3	86.2	84.5	76.1	75.3	80.5	3.01 M	8.1	90
+PEFNet+GDSFPN	86.1	83.3	78.3	77.6	81.3	3.34 M	8.9	65
+PEFNet+SDCLD	87.9	83.2	76.3	77.4	81.2	2.44 M	7.1	69
+GDSFPN+SDCLD	86.2	83.5	78.2	78.6	81.6	2.62 M	6.7	89
+PEFNet+GDSFPN+SDCLD	86.2	85.0	77.6	78.6	81.8	2.70 M	7.3	68
PGS-YOLO	87.1	85.6	78.1	78.5	82.3	2.70 M	7.3	69

**Table 9 sensors-25-04443-t009:** Ablation Experiment (Model Comparison).

Model	P%	R%	mAP50%	Params (M)	F1-Score	FPS	GFLOPs
Faster R-CNN [2]	45.3	72.6	71.1	28.31	0.60	15	108.0
SSD [5]	80.1	46.9	70.1	26.28	0.61	70	93.6
YOLOv3-tiny [18]	75.8	70.6	75.7	12.13	0.73	135	18.9
YOLOv5n [19]	80.8	67.9	77.6	2.50	0.74	69	7.1
YOLOv5s+Ghost+CA [6]	81.3	73.9	81.0	3.73	0.77	72	8.2
LE-YOLOv5 [7]	81.3	74.1	81.8	3.44	0.76	80	7.1
YOLOv6n [20]	79.5	70.9	76.8	4.23	0.75	81	11.8
YOLOv7-Tiny [21]	79.1	70.5	77.7	6.02	0.75	73	13.2
YOLOv8n [22]	81.1	72.8	79.4	3.01	0.77	92	8.1
YOLOv9t [23]	78.5	71.7	77.8	1.97	0.75	37	7.6
YOLOv10n [24]	78.7	69.8	77.7	2.27	0.74	71	6.5
YOLOv11n [25]	79.6	73.7	78.6	2.60	0.76	63	6.3
RT-DETR [26]	81.0	70.5	80.4	19.9	0.77	41	57.0
PGS-YOLO	81.8	74.5	82.3	2.7	0.78	69	7.3

**Table 10 sensors-25-04443-t010:** Comparison Experiment on the RDD2022 Full Dataset.

Model	P%	R%	mAP50	Para (M)	F1-Score	FPS	GFLOPs
YOLOv3-tiny	59.7	48.2	50.9	12.13	0.53	128	18.9
YOLOv5n	61.4	51.9	55.3	2.5	0.56	79	7.1
YOLOv7-Tiny	62.3	53.5	56.2	6.02	0.56	75	13.2
YOLOv8n	61.9	51.1	55.2	3.01	0.57	73	8.1
YOLOv10n	61.2	50.0	54.7	2.27	0.55	85	6.5
YOLOv11	63.1	52.9	57.1	2.58	0.57	44	6.3
RT-DETR	62.7	54.3	57.4	19.88	0.57	36	57
PGS-YOLO	63.2	54.5	58.0	2.70	0.58	64	7.3

**Table 11 sensors-25-04443-t011:** Comparison Experiment on the CRACK500 Dataset.

Model	P%	R%	mAP50	Para (M)	F1-Score	FPS	GFLOPs
YOLOv3-tiny	55.2	32.5	38.6	12.13	0.41	103	18.9
YOLOv5n	58.4	35.6	34.3	2.5	0.44	81	7.1
YOLOv7-Tiny	63.5	35.8	43.1	6.02	0.44	72	13.2
YOLOv8n	67.2	38.5	42.9	3.01	0.49	75	8.1
YOLOv10n	66.8	37.9	42.3	2.27	0.47	86	6.5
YOLOv11	65.7	37.4	41.8	2.58	0.46	58	6.3
RT-DETR	67.5	38.3	44.1	19.88	0.52	38	57
PGS-YOLO	68.0	38.8	45.2	2.70	0.51	72	7.3

## Data Availability

The open access dataset GRDDC2022 can be found at https://github.com/sekilab/RoadDamageDetector.git (accessed on 12 August 2022). The open access dataset CRACK500 can be found at https://github.com/guoguolord/CrackDataset (accessed on 15 May 2021).

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
