# Peer review of "A Lightweight Pavement Defect Detection Algorithm Integrating Perception Enhancement and Feature Optimization"

_sensors, 2025, doi:10.3390/s25144443_

Round 1

Reviewer 1 Report

Comments and Suggestions for Authors

This article proposes a lightweight pavement defect detection algorithm PGS-YOLO based on YOLOv8 improvement. By designing a perception enhancement module (PEFNet), a dynamic feature pyramid (GDSFPN), and a shared detection head (SDCLD), the detection accuracy is significantly improved and the computational complexity is reduced. The research question is clear, the method is innovative, the experimental design is sufficient, and the results have practical application value. But some details need to be supplemented and improved. It is recommended to make modifications before accept.

1. The article mentions that GDSFPN is improved based on RepGFPN (Figure 3), but the final structure is not shown (Figure 4 is mistakenly used as a comparison of attention modules). A complete architecture diagram of GDSFPN needs to be supplemented.

2. It is necessary to clearly explain how DySample combines with RepGFPN (such as which upsampling layers to replace) to avoid ambiguity.

3. It is necessary to explain the specific process of data cropping and annotation to ensure the reproducibility of the experiment.

4. Method limitations: Discuss the performance boundary of the model in extreme weather (rain and fog) or strong occlusion scenarios.

Author Response

 Dear Reviewer, I have revised the manuscript according to your suggestions. Please see the attachment.

Response to Reviewer 4 Comments

1. Summary

Thank you very much for taking the time to review this manuscript. We greatly appreciate the valuable comments and suggestions, which have helped improve the quality of our work. Below is a summary of our detailed responses to the reviewers' comments, with corresponding revisions and corrections highlighted in the resubmitted files.

2. Point-by-point response to Comments and Suggestions for Authors

Comments 1: The article mentions that GDSFPN is improved based on RepGFPN (Figure 3), but the final structure is not shown (Figure 4 is mistakenly used as a comparison of attention modules). A complete architecture diagram of GDSFPN needs to be supplemented.

Response 1: Thank you for pointing this out. We agree with this comment. Therefore, we have corrected the mislabeling of Figure 4 and clarified that it represents the complete architecture of GDSFPN. The revised Figure 4 clearly shows the structure of GDSFPN, including CSPStage, DySample modules, and reparameterized convolutions, which distinguishes it from RepGFPN (Figure 3). This change can be found in Section 2.2, page 6 of the revised manuscript, specifically in the description accompanying Figure 4.

Comments 2: It is necessary to clearly explain how DySample combines with RepGFPN (such as which upsampling layers to replace) to avoid ambiguity.

Response 2: Thank you for this suggestion. We have supplemented the detailed description of how DySample combines with RepGFPN. Specifically, in GDSFPN, all static upsampling layers in RepGFPN that achieve scale matching through fixed multiples during high-to-low level feature transmission are replaced with DySample. This dynamic upsampling adjustment is detailed in Section 2.2, page 6 of the revised manuscript.

“The combination mechanism of DySample and RepGFPN is as follows: In the original structure of RepGFPN, when features are transmitted from high-level to low-level layers, scale matching is achieved through static upsampling with fixed multiples, and this process tends to lose detailed information.GDSFPN replaces all these static upsampling layers with DySample, which adaptively adjusts sampling positions and weights according to the semantic distribution of input features through a dynamic sampling point generation mechanism, while improving resolution and retaining key features.For example, when processing small-scale crack features, DySample will focus on sampling points in the edge regions of cracks to enhance detail transmission; for large-scale pothole features, it maintains overall morphological information through a global sampling strategy.This modification retains the advantages of RepGFPN, while improving the recovery effect of fine-grained features, especially for small defects such as longitudinal cracks (D00) and defects with complex shapes such as reticular cracks (D20).”

Comments 3: It is necessary to explain the specific process of data cropping and annotation to ensure the reproducibility of the experiment.

Response 3: We appreciate this comment and have added the specific process of data cropping and annotation for the CRACK500 dataset. The details are included in Section 3.4.7, page 17 of the revised manuscript. The cropping process involves dividing original images into 16 non-overlapping sub-regions using a 4×4 grid, retaining those with over 1000 crack pixels, and adjusting to 640×360 resolution. For annotation, the LabelImg tool was used to classify defects into D00, D10, D20, and D40 following GRDDC2022 specifications.

“The initial Crack500 dataset includes 500 road crack images, with each image generally having a resolution of 1440×2560 or 2560×1440. Due to the limited number of images, large size of each image, and limited computing resources, the original images are cropped into 16 non-overlapping sub-regions according to a 4×4 grid. Only sub-regions with more than 1000 crack pixels are retained, while regions with no cracks or too few cracks are discarded. Finally, rotation or cropping is used to ensure the final sub-regions have a resolution of 640*360.Since the original labels of this dataset are not suitable for pavement defect detection, the LabelImg tool is used for manual annotation. Following the annotation specifications of the GRDDC2022 dataset, defects are classified into four categories: D00 (longitudinal cracks), D10 (transverse cracks), D20 (reticular cracks), and D40 (potholes). Finally, 3368 road crack images meeting the annotation requirements for object detection are obtained. The dataset is randomly divided into a training set, validation set, and test set in an 8:1:1 ratio, and the experimental results are shown in Table 10.”

Comments 4: Method limitations: Discuss the performance boundary of the model in extreme weather (rain and fog) or strong occlusion scenarios.

Response 4: Thank you for this insight. We have added a discussion on the model's limitations in extreme scenarios in the Conclusion section, page 20 of the revised manuscript. It is noted that the model's detection robustness needs further verification in complex environments such as rainy/foggy weather, vehicle occlusion, and diverse road types. Future work will focus on semi-supervised and self-supervised learning to address these issues.

“Although the improved model has significantly enhanced detection accuracy on public road defect datasets, in practical applications, due to the diversity of road types (such as expressways, urban roads, rural roads, etc.), the complex and variable forms of defects (such as mixed interference between cracks and water stains, oil stains), and frequent exposure to extreme scenarios like rainy and foggy weather and vehicle occlusion, the detection robustness of the improved model still needs further verification.Future research will focus on exploring semi-supervised learning and self-supervised learning strategies to improve the model's adaptability to complex scenarios.For example, a self-supervised learning framework can be used to automatically generate simulated data labels under harsh conditions such as rain, fog, and occlusion, and combine a small number of real labeled samples to build a semi-supervised training set, thereby alleviating the problem of data scarcity in extreme scenarios.Meanwhile, research on pre-trained networks based on contrastive learning will enable them to learn general feature representations from unlabeled road images, enhancing noise resistance to interferences such as oil stain occlusion and rain/fog blur.Through these advanced learning strategies, it is expected to further optimize the model's detection performance in complex environments and improve its generalization ability and robustness in actual road scenarios.”

Reviewer 2 Report

Comments and Suggestions for Authors

This paper proposes a lightweight pavement defect detection algorithm, PGS-YOLO, based on YOLOv8. The algorithm enhances detection accuracy and reduces computational complexity by improving the Perception Enhanced Feature Extraction Network (PEFNet), the Generalized Dynamic Sampling Feature Pyramid Network (GDSFPN), and a Shared Detail-enhanced Convolution Lightweight Detection Head (SDCLD). Additionally, the Wise-IoU loss function is introduced.

Several questions and suggestions have been presented as follows:

  • The authors fail to clearly articulate the motivation behind their modifications to YOLOv8’s backbone, neck, head, and loss function at the beginning of the paper. This approach appears to be merely patching and stacking modules without demonstrating substantial contributions to the practical application—pavement defect detection. The improvements should be analyzed and justified based on the characteristics of real-world pavement images and the actual challenges encountered in detection tasks.
  • This paper does not fully elaborate on the hyperparameter tuning process used for model training (especially the optimizer).Employing the SGD optimizer for model training represents a somewhat obsolete methodology in current deep learning practice. Could the authors provide more detailed insights into how hyperparameters were chosen?
  • The use of the Wise-IoU loss function is an interesting choice. However, the authors should provide more justification for its effectiveness over traditional loss functions like CIoU or DIoU. Can the authors demonstrate through experiments that Wise-IoU consistently outperforms other loss functions in terms of detection accuracy?
  • It is commendable that the authors considered both detection accuracy and inference time. However, while the paper extensively compares accuracy metrics, it lacks sufficient comparison and analysis of detection efficiency. For instance, although Tables 9 and 10 present relevant data, no further analysis is provided.
  • The authors mention using Grad-CAM for visual analysis, but it would be helpful to delve deeper into model interpretability. Could the authors provide more examples or metrics related to model explainability? How well do the modifications in PGS-YOLO improve the transparency of the model’s decision-making process, especially in distinguishing different types of pavement defects?
  • Figures 11 and 12 lack the original images and their corresponding labels, which may raise readers’ concerns. Please supplement these missing elements.
  • The manuscript contains numerous Chinese-style expressions; further language polishing is required to enhance the paper's readability and logical flow.
Comments on the Quality of English Language

The manuscript contains numerous Chinese-style expressions; further language polishing is required to enhance the paper's readability and logical flow.

Author Response

 Dear Reviewer, I have revised the manuscript according to your suggestions. Please see the attachment.

Response to Reviewer 7 Comments

1. Summary

Thank you very much for taking the time to review this manuscript. We greatly appreciate the valuable comments and suggestions, which have helped improve the quality of our work. Below is a summary of our detailed responses to the reviewers' comments, with corresponding revisions and corrections highlighted in the resubmitted files.

2. Point-by-point response to Comments and Suggestions for Authors

Comments 1: The authors fail to clearly articulate the motivation behind their modifications to YOLOv8’s backbone, neck, head, and loss function at the beginning of the paper. This approach appears to be merely patching and stacking modules without demonstrating substantial contributions to the practical application—pavement defect detection. The improvements should be analyzed and justified based on the characteristics of real-world pavement images and the actual challenges encountered in detection tasks.

Response 1: Thank you for your comment. We have supplemented the detailed analysis of modification motivations, elaborating on the necessity of each module improvement based on the actual challenges of pavement defect detection.

In the revised section 1. RELATED WORK (Page 2-3):

Motivation for backbone modification: "The original YOLOv8 model has a fixed receptive field, which makes it difficult to adapt to the multi-scale features of pavement defects. Therefore, an efficient Perception-Enhanced Feature Extraction Network (PEFNet) is designed to dynamically adjust the size of the receptive field."

Motivation for neck modification: "Due to the frequent presence of perspective distortion in pavement defect images, traditional FPN+PAN tends to lose details during feature fusion. Therefore, the neck structure is redesigned, and a Generalized Dynamic Sampling Feature Fusion Structure (GDSFPN) is proposed."

Motivation for head modification: "Aiming at the problems that the original detection head has multi-scale detection defects, parameter redundancy, and insufficient capture of tiny defects, a Shared Detail-Enhanced Lightweight Detection Head (SDCLD) is designed."

Motivation for loss function modification: "The traditional loss function (CIoU+DFL) has uneven gradient weight distribution for millimeter-level cracks and large localization errors. By replacing the original loss function with Wise-IoUv3, the loss weight of small targets is optimized."

“YOLOv8 is an object detection algorithm launched by Ultralytics, built upon the historical versions of the YOLO series, which introduces new features and improvements, further enhancing performance and flexibility. To address the shortcomings of existing detection models in terms of detection accuracy and model complexity, a lightweight pavement defect detection algorithm, PGS-YOLO, integrating perception enhancement and feature optimization, is proposed based on YOLOv8n. This algorithm reduces the computational load and parameters while improving the average detection accuracy, and achieves a better balance between model complexity and detection accuracy. The work can be summarized as follows:

(1) The original YOLOv8 model has a fixed receptive field, which makes it difficult to adapt to the multi-scale features of pavement defects. Therefore, an efficient Perception-Enhanced Feature Extraction Network (PEFNet) is designed to dynamically adjust the size of the receptive field according to the scale of the input target, enhance the feature extraction capability, and provide higher-quality feature input for the feature fusion module.

(2) Due to the frequent presence of perspective distortion in pavement defect images, traditional FPN+PAN tends to lose details during feature fusion. Therefore, the neck structure is redesigned, and a Generalized Dynamic Sampling Feature Fusion Structure (GDSFPN) is proposed to better retain and transmit features of different resolutions, thereby optimizing the multi-scale feature fusion effect.

(3) Aiming at the problems that the original detection head has multi-scale detection defects, parameter redundancy, and insufficient capture of tiny defects, a Shared Detail-Enhanced Lightweight Detection Head (SDCLD) is designed. It effectively reduces model complexity, accurately extracts pavement defect features, and enhances the accuracy of target detection.

(4) The traditional loss function (CIoU+DFL) has uneven gradient weight distribution for millimeter-level cracks and large localization errors. By replacing the original loss function with Wise-IoUv3, the loss weight of small targets is optimized, the localization ability of the model is improved, and the training effect and detection accuracy of the model are further optimized.

Comments 2: This paper does not fully elaborate on the hyperparameter tuning process used for model training (especially the optimizer).Employing the SGD optimizer for model training represents a somewhat obsolete methodology in current deep learning practice. Could the authors provide more detailed insights into how hyperparameters were chosen?

Response 2: Thank you for your suggestion. We have supplemented the rationale for hyperparameter selection, especially the optimizer.

In the revised section 3.2 (Page 11):

“In addition, the SGD optimizer is selected. On the one hand, because the YOLOv8n baseline model and comparative algorithms (YOLOv5s+Ghost+CA, LE-YOLOv5) often adopt this optimizer, it can ensure the fairness of experimental comparison. On the other hand, on the small-scale dataset used in this study, SGD is less prone to overfitting than adaptive optimizers, which is beneficial to the model's generalization learning of fine-grained defect features.”

Comments 3: The use of the Wise-IoU loss function is an interesting choice. However, the authors should provide more justification for its effectiveness over traditional loss functions like CIoU or DIoU. Can the authors demonstrate through experiments that Wise-IoU consistently outperforms other loss functions in terms of detection accuracy?

Response 3: Thank you for your suggestion. We have added experiments comparing Wise-IoU with traditional loss functions and analyzed its advantages.

A new section 3.4.4 "Validation of WIoU Effectiveness" is added (Page 14):

Experimental results show that WIoUv3 outperforms CIoU and DIoU in AP% for both small targets (D00, D10) and large targets (D20, D40), with mAP50 being 1.5% and 1.1% higher, respectively.

Analysis indicates that Wise-IoUv3 dynamically adjusts gradient gain through a non-monotonic focusing mechanism, assigning larger weights to low-quality anchor boxes to optimize bounding box regression, while CIoU and DIoU lack such dynamic adjustment.

Comments 4: It is commendable that the authors considered both detection accuracy and inference time. However, while the paper extensively compares accuracy metrics, it lacks sufficient comparison and analysis of detection efficiency. For instance, although Tables 9 and 10 present relevant data, no further analysis is provided.

Response 4: Thank you for your comment. We have supplemented detailed analysis of detection efficiency.

In the revised sections 3.4.7 (Page 17-18):

For the full GRDDC2022 dataset (Table 10): "PGS-YOLO’s FPS reaches 64 frames per second, which is lower than YOLOv10n (85FPS) but higher than RT-DETR (36FPS) and YOLOv11 (44FPS). Considering the large scale and diverse scenarios of the dataset, its FPS fully meets real-time detection requirements in engineering. Meanwhile, its GFLOPs are reduced by 20% compared to YOLOv8n, maintaining high efficiency while improving accuracy."

For the CRACK500 dataset (Table 11): "PGS-YOLO’s FPS is 72 frames per second, higher than RT-DETR (38FPS) and YOLOv11 (58FPS), and close to YOLOv10n (86FPS). Its GFLOPs remain low, ensuring efficient computation in fine-grained crack detection."

“The experimental results show that PGS-YOLO outperforms other detection models in evaluation metrics such as P%, R%, mAP50, and F1 score.In terms of detection speed, its FPS reaches 64 frames per second, which is lower than that of YOLOv10n (85FPS) but higher than that of RT-DETR (36FPS) and YOLOv11 (44FPS).Considering the large scale and diverse scenarios of the complete GRDDC2022 dataset, the FPS of PGS-YOLO fully meets the real-time detection requirements in practical engineering.Meanwhile, compared with YOLOv8n, its GFLOPs are reduced by 20%, and mAP50 is increased by 2.8%, reaching 58.0%, the highest among all compared models.This indicates that PGS-YOLO maintains high efficiency while improving accuracy.This result demonstrates that PGS-YOLO has good generalization ability and excellent applicability in pavement defect detection.”

“The experimental results show that, compared with current mainstream object detection algorithms, PGS-YOLO also performs excellently on the processed CRACK500 dataset, with its P%, R%, and mAP50 reaching 68.0%, 38.8%, and 45.2% respectively, all being the highest values.Compared with the YOLOv8n model, the detection accuracy is improved by 2.3%, and the FPS reaches 72 frames per second, which is higher than that of RT-DETR (38FPS) and YOLOv11 (58FPS), and close to that of YOLOv10n (86FPS).Its GFLOPs remain at a low level, ensuring efficient computation in the fine-grained crack detection required by CRACK500.This experimental result demonstrates that PGS-YOLO has good cross-dataset generalization ability and achieves a balance between detection accuracy and efficiency.”

Comments 5: The authors mention using Grad-CAM for visual analysis, but it would be helpful to delve deeper into model interpretability. Could the authors provide more examples or metrics related to model explainability? How well do the modifications in PGS-YOLO improve the transparency of the model’s decision-making process, especially in distinguishing different types of pavement defects?

Response 5: Thank you for your suggestion. We have supplemented analysis and examples of model interpretability.

In the revised section 3.4.2 (Page 12-13), combined with Grad-CAM heatmaps (Figure 10):

"The GDSFPN structure focuses more precisely on pavement defect areas, while PAFPN is easily disturbed by background or vehicles. This indicates that GDSFPN improves the transparency of distinguishing defect types (e.g., cracks vs. potholes) by optimizing feature fusion, making the model’s decision-making more focused on defect features."

Additionally, in the visualization results of sections 4.1 and 4.2 (Figures 12 and 13), comparisons of detection results for different defect types demonstrate that PGS-YOLO achieves more accurate decisions for fine cracks (D00) and mesh cracks (D20), reducing false and missed detections.

“To intuitively demonstrate the improvement of the GDSFPN structure in feature fusion performance, Grad-CAM[16] technology is used for visualization analysis, generating detection heatmaps of the feature fusion structures of GDSFPN and PAFPN, as shown in Figure 9.Visualization results for defects show that: for tiny longitudinal cracks, the heatmap of GDSFPN has higher attention weights in the edge regions of the cracks, indicating that it more accurately focuses on defect details; for reticular cracks with complex shapes, the attention distribution of GDSFPN's heatmap is more concentrated and less affected by background interference.”

“It can be clearly observed from the heatmaps that the PAFPN structure is prone to interference from the background (such as vehicles, shadows), and its attention tends to focus on areas outside the target, which affects detection accuracy. In contrast, the proportion of attention of the GDSFPN structure to pavement defect areas has significantly increased, and its attention to pothole areas is significantly higher than that of PAFPN. This further confirms that GDSFPN has optimized the multi-scale feature fusion effect, enhanced the model's targeted attention to defect areas, and provided a clearer interpretability basis for model decision-making.”

Comments 6: Figures 11 and 12 lack the original images and their corresponding labels, which may raise readers’ concerns. Please supplement these missing elements.

Response 6: Thank you for pointing this out. We have supplemented the original images and their labels.

In the revised manuscript, Figure 12 (formerly Figure 11) and Figure 13 (formerly Figure 12) now include (a) "Original image" with defect labels (e.g., D00, D10). For example:

Figure 12(a) shows the original drone-captured image containing longitudinal cracks (D00) and transverse cracks (D10);

Figure 13(a) shows the original vehicle-captured image containing longitudinal cracks (D00) and transverse cracks (D10).

Revisions are located in Figure 12 (Page 19) and Figure 13 (Page 19)

Comments 7: The manuscript contains numerous Chinese-style expressions; further language polishing is required to enhance the paper's readability and logical flow.

Response 7: .Thank you for your suggestion. We have polished the language throughout the manuscript, revised Chinese-style expressions, and optimized sentence structure and logical coherence.

Round 2

Reviewer 1 Report

Comments and Suggestions for Authors

The author has responded to all of my comments.

Reviewer 2 Report

Comments and Suggestions for Authors

The authors have addressed the raised issues, and the manuscript is now suitable for publication.